# Multiple Vaccines and Strategies for Pandemic Preparedness of Avian Influenza Virus

**DOI:** 10.3390/v15081694

**Published:** 2023-08-04

**Authors:** Hai Xu, Shanyuan Zhu, Roshini Govinden, Hafizah Y. Chenia

**Affiliations:** 1Jiangsu Key Laboratory for High-Tech Research and Development of Veterinary Biopharmaceuticals, Jiangsu Agri-Animal Husbandry Vocational College, Taizhou 225300, China; hai_x@126.com; 2Discipline of Microbiology, School of Life Sciences, College of Agriculture, Engineering and Science, University of KwaZulu-Natal, Durban 4001, South Africa; govindenr@ukzn.ac.za

**Keywords:** avian influenza virus, prevention, vaccine, delivery system, immune response

## Abstract

Avian influenza viruses (AIV) are a continuous cause of concern due to their pandemic potential and devasting effects on poultry, birds, and human health. The low pathogenic avian influenza virus has the potential to evolve into a highly pathogenic avian influenza virus, resulting in its rapid spread and significant outbreaks in poultry. Over the years, a wide array of traditional and novel strategies has been implemented to prevent the transmission of AIV in poultry. Mass vaccination is still an economical and effective approach to establish immune protection against clinical virus infection. At present, some AIV vaccines have been licensed for large-scale production and use in the poultry industry; however, other new types of AIV vaccines are currently under research and development. In this review, we assess the recent progress surrounding the various types of AIV vaccines, which are based on the classical and next-generation platforms. Additionally, the delivery systems for nucleic acid vaccines are discussed, since these vaccines have attracted significant attention following their significant role in the fight against COVID-19. We also provide a general introduction to the dendritic targeting strategy, which can be used to enhance the immune efficiency of AIV vaccines. This review may be beneficial for the avian influenza research community, providing ideas for the design and development of new AIV vaccines.

## 1. Introduction

Avian influenza virus (AIV) can infect not only birds but also other animals such as swine, cats, dogs, and even humans. The genome of AIV comprises eight gene segments: hemagglutinin (HA), neuraminidase (NA), matrix (M), nucleoprotein (NP), nonstructural protein (NS), acidic polymerase (PA), basic polymerase 1 (PB1), and basic polymerase 2 (PB2). Influenza viruses are differentiated into different subtypes based on the antigenicity of their HA and NA proteins, [1]. Currently, 16 HA subtypes and 9 NA subtypes have been identified in avian species. The AIV H9, H5, and H7 subtypes were first identified in China in 1994, 1996, and 2013, respectively. Influenza viruses circulating in animals have jumped to humans on multiple occasions [1], and some have demonstrated pandemic potential [2]. AIV infections have resulted in increased socioeconomic damages to humans as well as the poultry industry [3], especially the H5 subtype AIV, which is associated with higher mortality rates. The highly pathogenic AIV (HPAIV) is a bird-oriented virus with high morbidity and mortality, which leads to death of both poultry and humans. In addition, low-pathogenic AIV (LPAIV) strains do not directly affect poultry and healthy humans after infection, but uncontrolled and persistent poultry infection is not without consequences. For example, the LPAIV H9N2 subtype, which is highly prevalent in poultry, can be associated with morbidity and mortality when confounding factors are presented. Of grave concern, this virus might horizontally transfer internal genes to HPAIV, thus contributing to enhanced virulence and pathogenicity [4]. In China, which is recognized as a geographical area with conducive conditions to novel influenza virus emergence due to the location of three migratory bird flyways and the largest global scale of poultry production, at least six subtypes of AIV were identified in the period of 2016–2019 [5].

The Chinese government has invested significant resources into strengthening veterinary administration, education, research, investigation, surveillance, emergency responses, international cooperation, mass vaccination, and biosecurity improvements for AIV control [6]. Among these strategies, mass vaccination is still one of the most effective ways to control avian influenza outbreaks and for the alleviation of severe symptoms caused by AIV. Due to the complications associated with AIV epidemics, both culling and biosecurity measures have been strictly enforced in China, while mass poultry vaccination has been utilized as a rational alternative strategy to rapidly alleviate the severe situation [6]. The Chinese government has provided free AIV vaccines to poultry farmers. The mass vaccination strategy implemented in China has made great progress and has achieved satisfactory results. Consequently, the H5 clade 7.2, which circulated widely among dunghill poultry in northern China from 2006 to 2013, has been largely eliminated, as this clade has not been detected since the corresponding vaccine was used in 2014 [7].

Although the mass vaccination strategy achieves impressive results, immune escape happens occasionally; thus, AIV epidemics have not been totally controlled [8]. Scientists are thus concerned that high antibody levels can accelerate viral mutation and diversification under immune pressure, resulting in a more complicated epidemic situation in the long term [9]. The HA protein plays a critical role in virus binding to host cell surface receptors, thus mediating fusion of the host cell with viral membranes. Antibodies against HA, however, block this process by interfering with the binding between viruses and receptors, thus further blocking the downstream process [10]. The NA protein prevents the accumulation of influenza virus particles on the cell surface and is involved in virus release, thus facilitating further infection [11]. Both the HA and NA proteins are immunogenic, inducing a humoral immune response in infected animals. These proteins vary the most between viruses and are important for antigen drift, i.e., the gradual accumulation of amino acid changes that eventually reduce antibody recognition and immune escape [12]. Consequently, research efforts have focused on the improvement of vaccines and the development of new vaccine platforms to achieve substantial protection against potentially pandemic AIV.

## 2. Avian Influenza Vaccines

The development of a fully effective vaccine that can prevent AIV infection completely is a challenging task. An ideal avian influenza vaccine should meet the following requirements: inexpensive for production and low-cost for mass application, usable in multiple avian species, allows easy identification between infected birds and vaccinated population, be antigenically close to epidemic virus strain, provides long lasting protection after a single dose, induces a protective immune response in the presence of maternal antibodies and be applied at one day of age in a hatchery or in vivo [13]. Although various forms of vaccines have been developed to protect against influenza viruses, including traditional and next-generation vaccines (Figure 1), no current vaccine meets all these criteria. Therefore, the user must select licensed vaccines that fulfil as many of the features that satisfy their requirements. Currently, there are several licensed AIV vaccines, and some are under development in China; however, each has advantages and drawbacks (Figure 1).

Amongst these vaccines, the inactivated AIV are the most widely used in China. Due to immune escape caused by antigenic drift for the human influenza virus, the antigen of the vaccines must be replaced periodically to ensure that their antigenicity matches that of the current, circulating virus [5,9]. As for avian vaccines, the problems are mainly due to the fact, that there are viruses belonging to different subtypes, clades and subclades circulating among the bird population. Antigenic drift remains a concern for subtype-specific avian influenza vaccines. Therefore, antibodies to certain circulating viruses fail to provide protection against other viruses. Although antigenic drift of AIV can be predicted and the infrastructure and schedule to produce inactivated AIV vaccines is well established worldwide, this repetitive work needs to be done continuously. Therefore, the modification and innovation of AI vaccines is an ongoing subject that must learn from the human influenza vaccine research experience, including the improvement of immunogenicity, promotion of cross protection, and development of novel vaccine types or universal vaccines [14].

### 2.1. Inactivated Vaccines

The inactivated AIV vaccine is produced by growing the seed virus in chicken embryonated eggs, which is the most popular avian virus vaccine production technology process. The efficiency of this method, however, is relatively low, and requires a considerably large quantity of fertile eggs to prepare sufficient antigen [15]. The HA protein is an attractive target for vaccine development, with the bulky and highly variable-immunodominant globular head domains it presents being recognized by the host immune system. Previous studies have shown that the passage of seed virus in eggs may alter HA antigenicity, resulting in an antigenic mismatch with the epidemic isolates, thus making the vaccine less efficacious [16]. To avoid HA mutations caused by passage in eggs, cultured mammalian cell lines can be used for virus propagation [17]. Moreover, mammalian, cell-based influenza vaccines provide comparable or improved immune protection in animal models compared to egg-based vaccines, with both the safety and the efficacy having been proven by clinical trials [18,19]. The use of mammalian cell lines to culture the virus has several advantages, including the use of fully characterized and standardized cells and the ease of large-scale cultivation in the event of an emerging pandemic. In addition, the efficiency of the production process of cell-based vaccines is not only dependent on the ability of a particular cell to yield a virus vaccine, but also on the process optimization of large batches of virus vaccine. Currently, two immortal cell lines have been investigated for influenza virus production: Madin-Darby canine kidney (MDCK) cells and African green monkey kidney cells (Vero) [20,21].

The harvested and purified virions are inactivated and then emulsified with oil adjuvant. Formaldehyde is frequently used for inactivation to produce intact, inactivated vaccines [22], as it can interact with a broad range of reactive groups, whether protein, RNA, or DNA, leading to alkylating and homo- or bifunctional cross-linking. However, some disadvantages associated with the use of formaldehyde have been identified: one is the toxicity caused by residual formaldehyde in the vaccine, while another is the alteration of the antigen epitope due to chemical modification by formaldehyde [23,24]. Another commonly used inactivating agent for vaccine production is β-propiolactone. It functions as an alkylating agent; however, acylation and cross-linking of macromolecules may also occur during inactivation [25]. β-propiolactone can pass through the viral membrane, leading to an irreversible alkylation of nucleic acid bases, resulting in viral genome replication inhibition or genome degradation. Hence, β-propiolactone inactivates viruses via the denaturation of nucleic acids, in contrast to protein manipulation by formaldehyde. Thus, the antigenic structure of the virus will be maintained during inactivation by β-propiolactone. Researchers have investigated immunization with formaldehyde- and β-propiolactone-inactivated vaccines. It was observed that immunization with a β-propiolactone-inactivated H5N1 vaccine induced a higher level of cytotoxic CD8^+^ T cell and cytokine production [26]. Virus inactivation is also an important aspect of creating vaccines produced using reverse genetics technology. The segmented genome of AIV enables the rescue of viruses that contain gene segments encoding internal proteins of non-pathogenic AIV while displaying the surface proteins of highly pathogenic epidemic virus strains. Reverse genetics rescue can also be used to modify HA sequences to improve their replication capabilities in chicken embryos or cell lines to further enhance the efficiency of vaccine production [27]. Many inactivated reverse-genetics-designed AIV vaccines for chickens have recently been licensed in China (Table 1). Poultry immunization with inactivated virus vaccines begins at 2–5 weeks of age, with a booster immunization needed four months after the first immunization because the immunity conferred by the vaccine is not long-lasting. Researchers have demonstrated that injection with an inactivated AIV vaccine resulted in post-hatch seroconversion and the induction of immune responses in hatched chicks [28].

### 2.2. Live Vaccines

Live-attenuated vaccines for humans are available in the United States, Canada, and several European countries [30]. Vaccines derived from cold-adapted and temperature-sensitive master donor viruses [31,32] are propagated in eggs, causing egg-adaptive mutations in HA. A live-attenuated influenza virus administered intranasally would replicate in the nasal mucosa and induce immunity but would be attenuated because of its restricted ability to replicate in the lungs. Live-attenuated vaccines mimic the natural infection process; they can induce both IgA and IgG antibodies, without causing serious adverse reactions [33]. IgA, the principal isotype in secretions at the mucous membrane, can be detected on epithelial surfaces and in the upper respiratory tract. IgG, the principal isotype in blood and extracellular fluids, provides a cross-reactive immune response at the initial replication site [34,35]. There is no commercial live attenuated vaccine for avian influenza in China [36]. However, researchers have used live virus vectors to express the HA gene or combine the neuraminidase (NA) gene to develop AIV vaccines. Several virus vectors have been tried, viz., fowlpox-, Newcastle disease virus (NDV)- and herpesvirus-based AIV vaccines, which have been licensed in at least one country [37].

Fowlpox virus has been utilized as a vector for highly pathogenic AIV vaccines in research laboratories and in poultry farms in worldwide [38,39]. The fowlpox virus was used to express the HA and NA genes from the A/goose/Guangdong/1/1996 (H5N1) virus as a live vaccine, and its efficacy was proven in both laboratory and field tests [40]. This vaccine could protect against both a H5N1 and H7N1 virus challenge, indicating that the N1 component of the vaccine was able to protect against heterologous H7 virus [41]. If the chickens have, however, been previously exposed to live fowlpox virus or have the maternal antibody to fowlpox virus, then replication of fowlpox-AI virus and subsequent immune response will be reduced or blocked. This drawback greatly hampers the potential use of fowlpox-AI vaccine for chickens once they have been raised in the field [42].

Newcastle disease virus (NDV) has also been used as a vaccine vector and presents several advantages, including ease of preparation, high production yield, and its ability to serve as a bivalent vaccine against both NDV and AIV [43]. By using the reverse genetics system and engineered recombinant NDV expressing the HA genes from the H5N1 virus, it has been demonstrated that this recombinant virus induced strong hemagglutinin inhibition (HI) antibody responses to NDV and to H5 AIV in chickens [44]. In China, the NDV-AI vaccine has been licensed, and ~4 billion doses of this vaccine were utilized during the first two years following its marketing [45].

The turkey herpesvirus (HTV) was used as a vaccine for Marek’s disease in the 1970s [46], and it was also used to express AIV genes for vaccine development [47]. The HTV vector seems to overcome maternal antibody interference because virus spread occurs primarily from cell to cell. Moreover, the HTV vector could persist in the host and achieve the highest immunity as late as 9 weeks after immunization [48]. Furthermore, this HTV-AIV vaccine also induces both humoral and cellular response, which provides good clinical protection even though the antibody titer towards the challenge virus was low. The HTV-AIV vaccines have been licensed by different manufacturers in more than three countries [49]. The use of NDV-AIV and HTV-AIV vaccines have, however, been limited due to the high level of parent antibody, especially in China where the NDV and HTV immunization have been incorporated into the immune procedure.

### 2.3. Virus-like Particle (VLP) Vaccines

Virus-like particle (VLP) vaccines have attracted much attention as potential candidates for AIV vaccine development [50]. VLPs are self-assembling and non-infectious particles which indicates a high safety level for these vaccines. These particles can be produced from different expression systems such as bacterial, yeast, insect, and animal cell lines and may be used as particle vehicles or antigen in vaccine development due to their immunogenic characteristics. Moreover, they have comparable characteristics to the original pathogen, such as similar size, repetitive surface geometry, and ability to stimulate antigen-presenting cells (APCs), especially dendritic cells (DCs) as well as to induce both the humoral and cellular immune response [51,52]. Thus, the immune system of the host can recognize the VLP vaccine in an equivalent manner to the intact, inactivated virus vaccine to promote a robust immune response.

VLP-based vaccines have been used extensively in the development of influenza vaccines and have achieved encouraging results in providing immune protection. Researchers have shown that the H5N1-, H3N2-, and H9N2-VLP vaccines composed of the three influenza virus proteins, including HA, NA, and matrix 1 (M1) can be expressed in insect cells and assembled into VLPs [53]. Hu et al. [54] created a bivalent H5+H7 VLP vaccine using a baculovirus expression system that expresses the HA, NA, and M1 proteins. Both the bivalent VLP vaccine and commercial inactivated vaccines induced effective immune responses, including the production of antibodies that inhibited hemagglutination, neutralized the virus, and targeted HA. The bivalent VLP vaccine demonstrated significant reduction in viral shedding and replication in chickens, comparable to the effects of the commercial inactivated vaccine. Additionally, the bivalent VLP vaccine outperformed the commercial vaccine in terms of reducing pulmonary lesions caused by H7N9 virus infection in chickens. Moreover, VLP composed of HA and M1 induced high antibody levels in immunized mice and ferrets and provided immune protection against lethal virus challenge [55].

### 2.4. Universal Vaccines

The HA and NA proteins of the influenza virus have high mutation rates, so the seed virus of vaccines is also required to change to match the circulating influenza virus. Therefore, it is important to develop a universal influenza vaccine that can induce a long-term immune response and provide protection that covers a wide range of different virus strains [56]. Several efforts undertaken for the development of universal vaccines depend on conserved protective epitopes [57]. Promising universal vaccine candidates include the extracellular matrix 2 (M2e), the HA stem, the receptor binding site (RBS) of HA, and some of the cytotoxic T lymphocyte (CTL) epitopes in M1 and nucleoprotein (NP) [58]. Matrix 2 (M2) protein serves as a proton channel that also determines the entry and egress of the virus particle. The 24 amino acids of the extracellular region of M2 protein act as an interesting vaccination target due to them being highly conserved across multiple influenza virus strains [59,60,61]. Antibodies against M2e do not block the virus entering target cells but prevent virus release [62]. Despite the attractiveness of M2e as a vaccine antigen, the M2e immunogenicity is low after natural infection [63]. To overcome this obstacle, innovative approaches have been developed to induce M2e-directed antibody responses. A peptide vaccine consisting of the M2e epitope coupled with a fibrillizing peptide produced via artificial synthesis can self-assemble into nanoparticles in physiological salt solutions [64]. This strategy has demonstrated that immunization with M2-based nanoparticles could induce immune protection against both homologous challenge with the influenza PR8 H1N1 virus and the highly pathogenic avian influenza H7N9 virus. The conservative and cross-protective properties of M2e are exciting traits for the improvement of current influenza vaccines. Although several forms of the M2e vaccine have been evaluated, no licensed M2e vaccine is currently available on the market. Hence, to address this challenge with M2e-based vaccines, combination with other conserved epitopes is under consideration [61]. Despite the high mutation rate of the HA protein, the RBS of HA is functionally conserved because this site is essential for influenza virus entry. Antibody against RBS has shown a high cross-neutralizing capability [65]. Moreover, the HA stem is also highly conserved, making it a valuable candidate for broadly protective immune responses [66]. Thus, a chimeric HA can be constructed by combining the RBS and HA stem epitopes, where the head of HA can be changed [66,67]. Furthermore, NP and M1 are conserved among influenza A viruses, but they are not suitable for antibody-induction in vaccine development due to their lack of exposure on the virus surface [68]. However, epitopes in these proteins are responsible for CTL, resulting in a broadly cross-reactive immune response [69].

### 2.5. DNA Vaccines

A typical DNA vaccine is an antigen-coding gene inserted into a non-replicative eukaryotic expression plasmid vector, which is delivered to the host via direct gene transfer [70]. Then, the host cell expresses the antigen protein encoded by the plasmid and is presented to immune cells via the major histocompatibility complex (MHC) pathway. The immune response induced by DNA vaccines is the Th1 type of immune response, in which cell-mediated immunity to the DNA vaccine is more prevalent than humoral immunity [71]. A previous study demonstrated that the DNA vaccine mechanism mimics the cellular pathogenesis of the virus [72]. Antigen protein is expressed and digested into smaller peptides by intracellular proteasomes. These peptides are subsequently presented by MHC class Ⅰ molecules to elicit the antigen specific CD8^+^ T cell responses. In addition, the peptides are also presented by MHC class Ⅱ molecules to activate CD4^+^ T helper cells, which trigger active B cells to produce antigen-specific antibodies [73].

Early studies have shown that the DNA vaccine-encoding HA gene can protect against lethal homologous challenge depending on an elevated level of HA-specific serum antibody [74]. A plasmid DNA dose of approximately 200–400 μg is required for an efficient DNA vaccine against highly pathogenic AIV challenge in chickens. This high dose requirement for immunity is a major obstacle to field use of such a vaccine. On the other hand, efficient antigen expression in the host is key for DNA vaccines. Based on the codon bias of the host species, the antigen coding sequence can be modulated to enhance the expression efficiency [75]. This strategy can decrease the DNA vaccine dosage through improved expression of the antigen gene. Animals immunized with a codon-optimised HA plasmid demonstrated 4-fold higher antibody titres compared to the animals immunized with the wild type HA plasmid, leading to a greater survival rate in viral challenge tests [76,77].

DNA vaccines demonstrate a multitude of desirable characteristics for influenza control and have undergone testing for various diseases, encompassing viral and bacterial infections as well as certain types of cancers [78,79]. In contrast to inactivated influenza vaccines, which primarily depend on the production of antibodies for effective protection [80], DNA vaccines have the ability to efficiently stimulate both humoral and cellular immune responses [81]. The preparation of DNA vaccines does not require the growth of live viruses and can be swiftly scaled up in response to emerging pandemic influenza situations [82,83]. Despite these advantages, the promising immunogenic responses achieved in small animal models, predominantly mice, are rarely replicated in larger animals [84]. Data from murine models are derived from immune responses in highly inbred animals that have been exposed to mouse-adapted influenza viruses. This provides an unreliable basis for comparison to vaccination outcomes in the outbred human population, where the goal is to protect against circulating influenza viruses [85]. Larger animal models, such as ferrets and cynomolgus macaques, offer more pertinent data as they are susceptible to human influenza viruses. Ferrets exhibit clinical signs, lung pathology, and transmission patterns similar to humans, while cynomolgus macaques demonstrate human-like immune responses to influenza, making them reliable predictors of vaccine efficacy in humans [86]. Therefore, the attainment of adequate immunogenicity in larger animals has required the development of potent delivery systems and adjuvants [87].

### 2.6. mRNA Vaccines

In 1993, Martinon et al. [88] conducted the pioneering study on the efficacy of a conventional mRNA vaccine against influenza. They observed that a cytotoxic T-cell response was induced in mice upon the administration of a liposomal vaccine encoding the influenza nucleoprotein (NP). Subsequently, other researchers demonstrated that injection of pigs, ferrets, and chicken with different mRNA encoding HA, NA, and NP elicited measurable immune responses [89].

Currently, mRNA vaccines can be divided into two major types: conventional, non-amplifying mRNA molecules and self-amplifying mRNA (saRNA) that maintain auto-replicative activity derived from an RNA virus vector [90]. Non-replicating mRNA vaccines can be produced via the incorporation of various modified nucleosides. Numerous studies have concentrated on the advancement of these vaccines specifically targeting influenza viruses. Pardi et al. [91] reported that immunization of mice and ferrets with a single 3 μg dose of an mRNA-LNP-HA vaccine induced NAbs titres > 1:120 four weeks post injection. A second dose increased these hemagglutination inhibition antibody titres to values of 1280–20,480, according to the dose and route of injection. Protective immune responses were observed in mice, ferrets, and pigs when they were intradermally injected with RNA vaccines encoding influenza HA, NP, and NA. Additionally, mice that received intravenous injection of PR8 H1N1 influenza A virus HA-encoded unmodified mRNA-lipid complex exhibited enhanced T-cell activation [92]. saRNA vaccines can be delivered in various forms, including virus-like RNA particles, plasmid DNA, and in vitro transcribed RNA. There have been multiple reports on the use of saRNA vaccines against influenza virus. Fleeton et al. [93] conducted a study where immunization with 10 mg of an saRNA vaccine encoding PR8 H1N1 influenza A virus HA resulted in a protective immune response against lethal homologous viral challenge in mice. Another study employed lipid nanoparticles to encapsulate saRNA-encoding HA, leading to the induction of protective levels of hemagglutination inhibition titres after two-dose intramuscular injections in mice [94].

## 3. Strategies for DNA Vaccine Delivery

Delivery strategies for DNA vaccines have been a hot area of research over the last decade [95]. An efficient DNA vaccine needs to be able to enter host cells and achieve protein expression; moreover, this vaccine must also be able to alert the immune system to its presence and induce an immune response. Kim et al. [96] indicated that following administration of a naked DNA vaccine, there is a rapid migration from the injection site coupled with plasmid DNA degradation; as a result, only occasional detection of plasmid DNA can be observed after 8 h in mice. Researchers agree that direct delivery of DNA to APCs offers a potential vaccine delivery system. However, effective delivery of a plasmid DNA vaccine into cellular nuclei requires the crossing of several barriers, including the phospholipid cellular membrane either through endocytosis or pinocytosis, confronting degradation in endosomes and lysosomes, surviving cytosolic nucleases, translocating across the nuclear envelope, and finally achieving antigen protein expression. So, the key point of DNA vaccine delivery systems is thus to overcome these biological barriers and target the immune cells and safely deliver plasmids to the nuclei of cells for protein expression. Several strategies have been attempted to improve the efficiency of DNA vaccine delivery, including polymers, liposomes, live bacteria, etc [97]. In addition, the vaccine potential of lipid nanoparticles (LNPs) has garnered attention for DNA vaccine delivery following COVID-19 vaccine development. Moreover, the gene gun, a mechanical delivery apparatus which can introduce macromolecules into the target cells, has also been used for DNA vaccine delivery.

### 3.1. Polymer Delivery Systems

Polymers have been widely used as delivery systems in applications such as tissue engineering, gene therapy, and DNA vaccination [98]. DNA material can be packaged by polymers into nano- and micro- particles to prevent damage by nucleases, and allow for tunable degradation and controlled release [99]. Moreover, this particle structure can be easily captured by the immune cells. Nowadays, various polymers have been explored for DNA vaccine development (Figure 2).

Poly (lactide-co-glycolide) (PLGA) has been used to package and deliver DNA vaccines against a variety of animal diseases, such as influenza [100], foot and mouth disease virus (FMDV) [101], and parasitic infections [102]. Packaging DNA vaccine into PLGA leads to an increased systemic, antigen-specific antibody and T cell proliferation response. In addition, DNA-coated PLGA microparticles have also been reported to enhance the delivery of DNA vaccine to APCs. However, although the delivery of DNA using PLGA has been found to induce immune responses, there are still several problems including DNA degradation during the encapsulation process and lower transgene expression due to the micron size. Therefore, researchers developed a modified PLGA nanoparticle using a glycol chitosan shell for dual live cell tracking and DNA vaccine delivery [103]. These particles can directly transfect Langerhans cells, with enhanced gene transcription and expression. Thus, the PLGA nanoparticles promote DNA migration to the lymph nodes and the activation of naïve B and T cells [103].

Poly-ethyleneimine (PEI) is another polymer that has been widely used in DNA vaccine delivery [104]. The use of PEI for DNA vaccine delivery is well studied and has been reported to improve the humoral response of a H1N1 DNA vaccine [100]. However, DNA/PEI complexes suffer from toxicity problems, aggregation in the presence of serum proteins, and rapid clearance from circulation, which results in a limited efficiency for DNA delivery. To overcome these issues, γ-polyglutamic acid was used to modify the DNA/PEI complex by reducing the surface charge of the complex, leading to decreased aggregation and stability in a physiological environment. γ-polyglutamic acid is produced by certain strains of bacilli, and it has been hypothesized to act as an adjuvant through interaction with the receptors of immune cells [105].

Some natural materials have also been investigated for other biological applications, including gene delivery due to their inherent biocompatibility, non-toxicity, biodegradability, stability, inexpensive production, and immune stimulation [106]. For instance, inulin, hyaluronic acid, alginate, as well as chitosan have attracted attention in the field of vaccine delivery [106]. Chitosan, the partially deacetylated form of chitin from shells, has been well studied for DNA vaccine delivery [107]. Chitosan must dissolve in a slightly acidic environment (pH < 5); this property makes it suitable for chemical alteration, thereby changing the solubility and charge and making it suitable for various applications. Additionally, chitosan has been identified as a non-toxic and biocompatible material, so its use has been approved by the Food and Drug Administration [108]. Plasmid DNA immobilized within chitosan-coated microspheres (20 to 50 μm) can induce both mucosal and systemic immune responses [109].

### 3.2. Liposome Delivery System

Liposomes were first identified and used to provide a model for the study of biological membranes in the 1960s. All the liposomes have a common character, i.e., they are amphiphilic molecules with cationic groups in the head. For continued research, the advantages of liposomes are their high loading capacity, biodegradability, higher safety, as well as comparatively easy and low-cost production [110]. Furthermore, liposomes can be designed according to the modular principle; thus, the structure, linker, and lipophilic area can be modified to obtain a higher delivery efficiency [111,112,113]. There are four key types of liposomes including conventional liposomes, polyethylene glycol-modified liposomes (PEGylated), ligand-targeted liposomes, and antibody-modified liposomes (Figure 3).

Conventional liposomes were the first generation of liposomes which consisted of a lipid bilayer composed of cationic, anionic, or neutral lipids. To enhance liposome stability and improve its circulation times in the organism, sterically stabilized liposomes were introduced via modification with polyethylene glycol [114]. Furthermore, ligand-targeted liposomes offer considerable potential for site-specific delivery of the DNA vaccine to designated cell types, especially APCs, which selectively express or over-express specific ligands on cell surfaces. Many types of ligands are available, including peptides, proteins, and small molecules. Antibodies, particularly monoclonal antibodies (mAb), nanoantibodies, and the antigen binding area of an antibody which can be affixed to the liposome surface, are one of the more versatile ligands [115].

Liposomes form spherical vesicles with a structure composed of phospholipids and cholesterol arranged into a lipid bilayer, allowing for fusion with cellular membranes [116]. Plasmid DNA can be sequestered either on the liposome bilayer or encapsulated within the liposomal vesicles. The application of liposomes as vehicles for DNA vaccine delivery has garnered significant interest, owing to their capacity to generate size-regulated entities, confer carrier customization, and elicit activation of innate immune receptors [117]. Liposomes possess the ability to generate submicron-scale particles that facilitate the incorporation of a substantial portion of DNA, thereby preventing the displacement of DNA due to anion competition. This characteristic ensures that a significant amount of DNA is effectively entrapped within the lipid bilayers [118].

In vaccine formulations, liposomes exhibit the ability to regulate the localized distribution within tissues, enhance retention at the site of injection, and modulate cell trafficking [119]. Vaccines formulated with liposomes offer several advantages, including the promotion of enhanced antibody production and cytotoxic T lymphocyte (CTL) responses. However, it is important to note that the specific characteristics of immune responses elicited by liposomes vary based on factors such as lipid composition, particle size, surface charge, and the location of entrapped antigens [110]. Studies have demonstrated that liposomes show promise as candidates for the delivery of DNA vaccines to mucosal tissue. Liu et al. [120] used liposomal systems to deliver a DNA vaccine encoding the influenza A virus M1 gene via oral administration which generated both humoral and cellular immune responses, accompanied by an elevation in IFN-γ production.

### 3.3. Live Bacteria Delivery Systems

The bacteria used for DNA vaccine delivery systems are recombinant bacteria that have been genetically modified, ensuring that majority of their pathogenicity components have been deleted, ensuring the safety of the host [121]. As a DNA vaccine delivery system, bacteria are divided into two major groups: non-pathogenic bacteria and attenuated, pathogenic bacteria. The attenuated bacteria that have been studied for DNA delivery include *L. monocytogenes* [122], *Salmonella* species [123], *Shigella* species [124], and *Yersinia enterocolitica* [125]. Pathogenic bacteria target the mucous membranes as their infection route; thus, they are suitable for mucosal administration. However, the main disadvantage is the probability of causing infection, particularly in infants and immunocompromised individuals [126]. Therefore, non-pathogenic bacteria such as lactic acid bacteria [127] may be preferable for development as a DNA vaccine delivery system.

If bacteria are used to deliver DNA vaccines, the vaccines can be delivered through mucosal routes, including intranasal and oral routes. Oral administration does not require special skills and is easier to manage, while administration via the intranasal route has hindered enzymatic reactions and can circumvent the high acidity conditions in the gut [128]. Following oral administration, bacteria with the DNA vaccine enter the digestion system, where they are recognized by the M cells in Peyer’s patches (Figure 4) and are transported through the intestinal surface to the lamina propia [128]. The expressed antigen in the host is presented by MHC I and activates the CD8^+^ T cells, and can also be expressed as extracellular protein, presented by MHC II, thus activating antibody production and T helper CD4^+^ cell responses [129]. Therefore, this route of administration induces both mucosal and systemic immune responses [123].

Lactic acid bacteria are excellent candidates to be engineered for DNA vaccine delivery [130]. They have been used in food fermentation since antiquity and are generally recognized as safe organisms [127]. Several lactic acid bacterial strains are acknowledged as probiotic bacteria which can increase the immune response toward pathogens by inhibiting pathogen colonization in the gastrointestinal tract and promote the mucosal immune system [131]. Yagnik et al. [132] reported that *Lactobacillus lactis* is capable of transferring plasmid DNA into Caco-2 cells in the absence of chemical treatment. *Salmonella* species are Gram-negative bacteria that cause salmonellosis through orofecal routes. Kong et al. [128] constructed a recombinant attenuated *Salmonella* mutant strain with a hyper-invasive phenotype that efficiently delivered DNA vaccines after entering the host cells. As a DNA vaccine delivery strategy, it is suitable for oral administration to simulate its natural infection route. Mutant *Salmonella typhi* and *Salmonella typhimurium* were developed as DNA vaccine delivery vehicles via *aroA*, *aroC*, or *aroD* gene mutations.

## 4. Dendritic Cell Targeting Enhances Vaccine Immune Efficiency

B and T lymphocytes are initiated into adaptive immune responses through the actions of APCs, which capture antigens from the internal environment and present them to these lymphocytes. APCs constitute a heterogeneous group of cells including macrophages, B cells, subsets of myeloid DCs, and plasmacytoid DCs (pDCs). Given the pivotal role of APCs in triggering immune responses, numerous strategies have been explored to specifically target antigens to one or multiple subsets of APCs. Of these various cell types, DCs have been recognized as particularly significant due to their highly efficient antigen uptake, processing, and subsequent presentation to T cells [133]. The exposure to targeting material leads to DC maturation and migration to the lymphoid node, which yields the immune response. Meanwhile, the DC-targeting material can also function by producing protective cytokines, such as IL-2, IL-4, and IFN-Ⅰ, which influence distinct steps in the adaptive immune response and the activation of innate lymphocytes [134,135]. Thus, targeting antigen delivery to DCs is a more direct and less laborious strategy to induce effective immune responses, which has been attracting considerable attention recently.

Cells of the innate immune system express pattern recognition receptors (PRRs), also referred to as pathogen recognition receptors or ancestral pattern recognition receptors due to their emergence prior to the development of adaptive immunity [136]. There are two main classifications of pattern recognition receptors (PRRs). The first group is comprised of endocytic PRRs that specifically bind to carbohydrates. This group includes the mannose receptor (MR), glucan receptor, and scavenger receptor (SR). The second group is composed of signaling PRRs, which encompass the membrane-bound, Toll-like receptors (TLRs) and the cytoplasmic NOD-like receptors. Numerous PRRs have been identified on the surface of DCs, including DC-SIGN, MR, TLR, SR, and DEC-205. Among these, the C-type lectins form a diverse family of lectins characterized by their carbohydrate recognition domains. In the context of DCs, key C-type lectins such as DC-SIGN, DC-SIGNR, DCAR, DCIR, Dectins, and DLEC play vital roles in processes such as trafficking, formation of the immunological synapse, and the initiation of both cellular and humoral immune responses [137]. TLRs are innate immune receptors that can use pattern recognition processing of ligands to detect a variety of molecules, including tissue damage signs, bacteria, virus-es, protozoans, and nematodes [138]. Thirteen known TLRs have been identified that can recognize a wide range of microbial antigens but differ in their specificity for microbial patterns. Targeting of these receptors is becoming an efficient way of delivering antigens in DC-targeted vaccines [139].

The utilization of the DC-targeting strategy is in the pipeline for the development of vaccines against viral pathogens and cancers (Figure 5). The antigens were designed to attach molecules that were identified to bind to DCs. When these antigen-carrying molecules are injected into the body and taken up by DCs, which then activate the immune system to mount a response. Jauregui-Zuniga et al. [135] described the development of a mAb against the carbohydrate recognition domain-2 (CRD-2) of the chicken DEC-205 receptor and then conjugated that with the purified HA to direct the antigen to the DCs, which elicits a strong immune response in chickens as early as 14 days after priming [135]. Gudjonsson et al. [136] showed that targeting influenza HA and chemokine receptor Xcr1^+^ to DCs induces immune responses and confers protection against the influenza virus. In addition, some studies have shown that adding a type of adjuvant called a TLR agonist to the influenza vaccine can increase the number of DCs that are activated, leading to a stronger and more durable immune response.

## 5. Conclusions

In summary, mass vaccination is still an essential part of a multi-pronged strategy to control AIV infection and transmission in poultry. The continuous threat of an AIV pandemic underscores the necessity to develop novel vaccines with a more streamlined and easy manufacturing process and effective immune protection than prevailing AIV vaccines against potential pandemic strains. Despite recent progress in the development and design of multiple of AIV vaccines against pandemic threats, several other points are still worthy of attention for developing an ideal AIV vaccine. The development of efficient adjuvants is an area of focus in AIV vaccine research. New adjuvants that can stimulate the immune system more effectively while minimizing side effects are being developed, which could improve vaccine efficacy and safety. Moreover, changing the route of immunization should not be neglected, since the use of intracutaneous injection has been shown to be more effective in generating immune responses and may require smaller doses of vaccine, potentially increasing vaccine availability. Overall, the multiple strategies presented in this review provide a better understanding of the current AIV vaccines. The future of vaccine development looks promising as researchers continue to explore innovative approaches to create more effective and accessible vaccines.

## Figures and Tables

**Figure 1 viruses-15-01694-f001:**
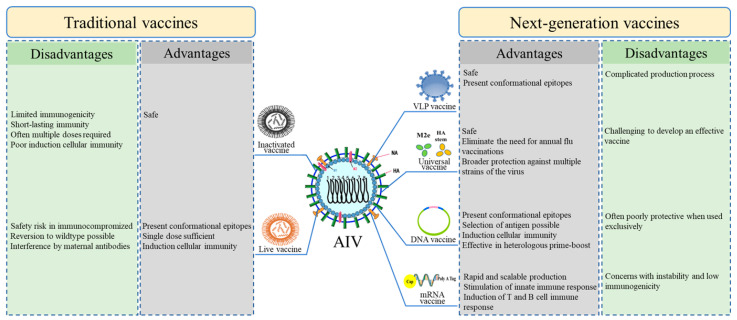
Advantages and disadvantages of traditional and next-generation influenza vaccines. The development of influenza virus vaccines is based on intact virus particles, surface proteins, nucleoproteins, viral genomes, and attenuated virus strategies.

**Figure 2 viruses-15-01694-f002:**
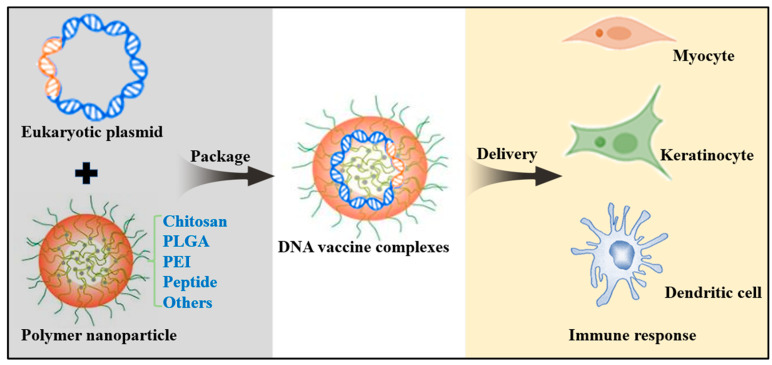
Polymer-based DNA vaccine delivery designs. Polymers can be made from monomer units obtained from living organisms or artificially synthesized, including chitosan, poly-lactide-co-glycolide (PLGA), poly-ethyleneimine (PEI), polypeptides, and other non-ionic block copolymers. Polymers can protect DNA from degradation through complexation or encapsulation, and delivering it to host cells to induce immune response.

**Figure 3 viruses-15-01694-f003:**
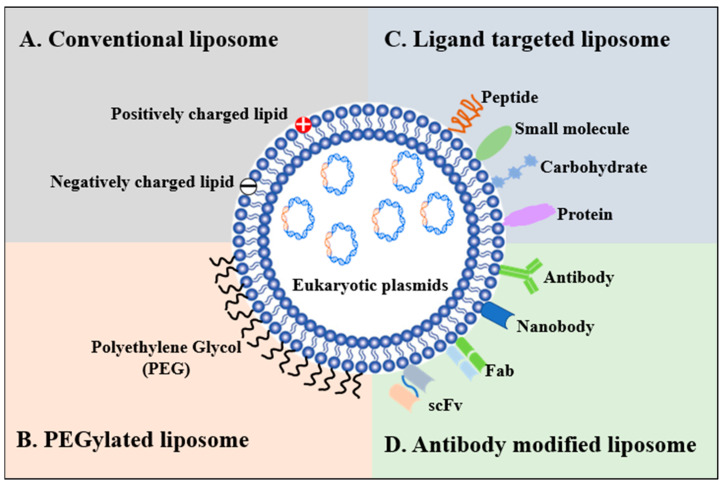
Schematic representation of the different types of liposomal delivery systems. (**A**). Conventional liposome: liposome consists of a lipid bilayer that can be composed of positive, negative, or neutral lipids, which encloses an aqueous core. (**B**). PEGylated liposome: the liposome surface can be modified via addition of polyethylene glycol to confer steric stabilization. (**C**). Ligand-targeted liposome: liposome surface can be modified bviaaddition of ligands for specific targeted delivery. (**D**). Antibody-modified liposome: antibody (monoclonal antibodies, nanobodies, Fab, and scFv) can be affixed to liposome surface.

**Figure 4 viruses-15-01694-f004:**
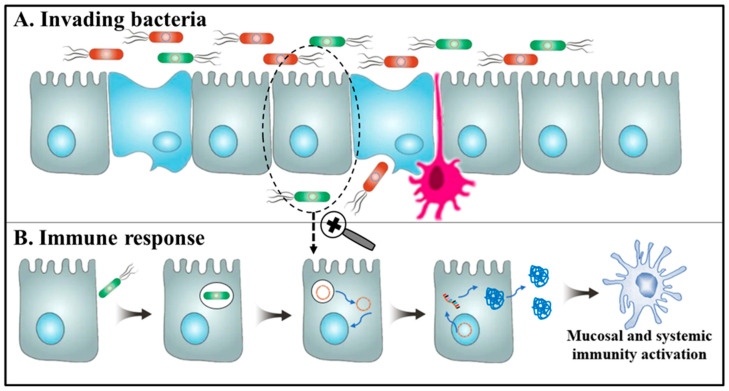
Schematic representation of the mechanism proposed for DNA vaccine delivery into mammalian cells using live bacteria. (**A**). Following invasion of the host, bacteria are recognized by various cell types, including microfold cells, epithelial cells, or immune cells. (**B**). The recombinant bacteria are internalized by the phagolysosome complex and subsequently undergo lysis. Within this process, the DNA vaccine is released from the vesicle and translocated into the nucleus, allowing for transcription of the antigen gene. The resulting antigen protein is presented to the immune system, triggering both cellular and humoral immune responses.

**Figure 5 viruses-15-01694-f005:**
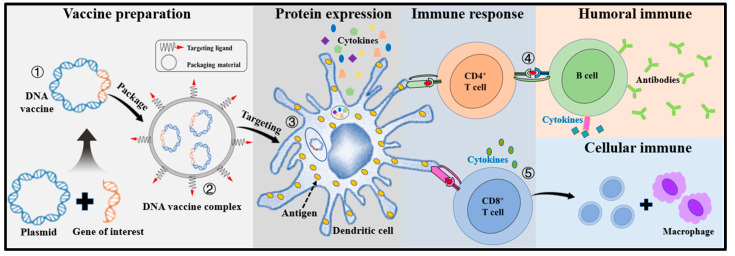
Introduction of dendritic-cell-targeting delivery strategy of DNA vaccine. Step 1: a DNA vaccine consists of a plasmid that encodes the antigen gene in the presence of a mammalian promoter. Step 2: it is encapsulated by packaging material which was surface modified with dendritic-cell (DC)-targeting ligands. Step 3: after targeting the DNA delivery vaccine complex to DCs, the encoding antigen is expressed inside the DC, and through the major histocompatibility complex (MHC) pathways to present the processed antigen to activate naïve T cells. Step 4: CD4^+^ T helper cell activation is triggered through MHC class Ⅱ pathway from DCs, and B cells will be activated by recognizing the secreted antigen or activated CD4^+^ T helper cell to produce different classes of antibodies. Step 5: CD8^+^ T cell immunity is predominantly activated by endogenously expressed antigens presented on MHC class Ⅰ molecules. The released cytokines (interferon-gamma [IFN-γ] or tumor necrosis factor-alpha [TFN-α] inhibit viral replication and enhance the expression of MHC Ⅰ molecules. Meanwhile, macrophages are also activated to support cell-mediated immune responses.

**Table 1 viruses-15-01694-t001:** Application periods of the H5 subtype AIV vaccine in China [15,29].

Seed of HA-Donor Virus	Clade	Designations	2004~ 2005	2006~ 2007	2008~ 2009	2010~ 2011	2012~ 2013	2014~ 2015	2016~ 2017	2018~ 2019	2020~ 2021	2022~ 2023
A/GS/GD/1/1996(H5N1)	0	Re-1	√	√	√							
A/CK/SX/2/2006(H5N1)	7.2	Re-4		√	√	√	√					
A/CK/AH/1/2006(H5N1)	2.3.4	Re-5			√	√	√					
A/DK/GD/S1311/2010(H5N1)	2.3.2	Re-6					√	√				
A/CK/LN/S4092/2011(H5N1)	7.2	Re-7						√	√			
A/CK/GZ/4/2013(H5N1)	2.3.4.4g	Re-8							√	√		
A/DK/GZ/S4184/2017(H5N6)	2.3.4.4h	Re-11								√	√	
A/CK/LN/SD007/2017(H5N1)	2.3.2.1f	Re-12								√	√	
A/DK/FJ/S1424/2020(H5N6)	2.3.4.4h	Re-13										√
A/WS/SX/4-1/2020(H5N8)	2.3.4.4b	Re-14										√

Note: HA stands for hemagglutinin. Abbreviations: GS—goose; CK—chicken; DK—duck; WS—whooper swan; GD—Guangdong; SX—Shanxi; AH—Anhui; LN—Liaoning; GZ—Guizhou; FJ—Fujian.

## Data Availability

The data presented in this study are available in the referenced articles.

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
