# Peer review of "Multiple Vaccines and Strategies for Pandemic Preparedness of Avian Influenza Virus"

_viruses, 2023, doi:10.3390/v15081694_

Round 1

Reviewer 1 Report

The paper by Hai Xu, Shanyuan Zhu, Roshini Govinden and Hafizah Y. Chenia entitled “Multiple vaccine strategies for pandemic preparedness of avian influenza virus in poultry” reviews the literature on different types of vaccines against avian influenza viruses of pandemic potential. The paper can be interesting for a wide range of experts in virology, molecular biology and biotechnology, as the paper is not so much an analytical review, as an introduction to general acquaintance with the topic. Furthermore, the problems related to the development of vaccines against influenza viruses of pandemic potential are insufficiently described in the manuscript. The differences in approaches to vaccines for poultry and human are absent in the paper. Thus, the paper requires major revision.

Title – as the paper mainly discusses influenza vaccines in general, the mention of poultry should be removed from the title.

47-50 “in 2016-2019” should be added.

61-63 It should be indicated that the viruses of that clade were circulating mainly among dunghill fowl in China.

64-76 It concerns mainly human influenza viruses, especially “The NA protein is involved in virus release and plays a role in breaking down the mucus barrier that protects the respiratory tract”. Avian influenza virus entries typically via digestive system.

97-99 Antigenic drift results in the main problems in developing human vaccines. As for vaccines for birds, the problems mainly caused by the fact, that among the bird population there are circulation of viruses belonging to different subtypes, clades and subclades. Therefore, antibodies to certain circulating viruses fail to provide protection against other viruses.

104 Reference [14] is devoted to seasonal vaccines for human immunization. The problems of human vaccines differ from that of vaccines for birds.

125 Reference [21] is about using VERO cells for rabies virus production.

144-146 The segmented genome of AIV allows for the rescue of viruses that contain internal gene segments of non-pathogenic AIV while displaying the surface proteins of highly pathogenic epidemic virus strains. – Incorrect statement, as all gene segments are within viral particle. Internal and external refer to viral proteins.

159 Live vaccines. Usually it is referred to attenuated vaccines. I recommend the authors should create special subsection on vaccines based on viral vectors.

225 Universal vaccine. This section is rather superfluous. Universal vaccines are rapidly developing trend, while the references belong to 2017-2018 and even to 2009 and 2011 years.

233-234 M2 protein as a whole serves as a proton channel, while M2e is an extracellular region of the influenza virus M2 protein.

264-265 A previous study demonstrated that the DNA vaccine mechanism mimics the cellular pathogenesis of the virus – incorrect statement. Pathogenesis is the process by which a disease or disorder develops

298 mRNA vaccines. The section 2.6 mRNA vaccines is almost entirely paraphrase of a part of the paper [92] - Reina, J., Enferm Infecc Microbiol Clin (Engl Ed) 2023. 41, 301-304.

303 Other researchers also demonstrated that injection of pigs, ferrets, and chicken with different mRNA encoding HA, NA and NP induced a detectable immune response [89] - Reference [89] is devoted only to the vaccine based on HA.

310-311 A second dose enhanced these antibody titres to values of 1280-20480 – Which approach was used to obtain these data? HI-test, MN, ELISA or another?

512 Please, change (Lu et al; 2022) with numerical reference.

525-527 One group of researchers targeted influenza M2e to DCs by fusing M2e with anti-Clec9, while HA of influenza was infused with an artificial adjuvant vector cell targeting DCs to induce CD4+ T cells and CD4+ Tfh cells [139] – The paper [139] discusses other topic.

Author Response

Pls see attachment

Reviewer 2 Report

“Multiple vaccine strategies for pandemic preparedness of avian influenza virus in poultry” by Hai Xu and all.

Authors wrote a review, about recent progress of AIV vaccines, which are based on the classical and next-generation platforms.

The information provided in the manuscript is relevant and interesting, because Avian Influenza viruses are still a big concern for animal and human health.

To write a review is not very easy, and you have to consider plagiarism.  The next paragraph between Lines 422-434 must be paraphrased.

 “Plasmid DNA can be either bound to the liposome surface or entrapped within the liposome vesicles. The use of liposomes for the delivery of DNA vaccines has attracted much attention due to the ability to form size controllable particles, functionalize the carrier, and activate innate immune receptors [117]. Liposomes are able to generate submicron-sized particles that can incorporate most of the DNA in a way that prevents DNA displacement through anion competition, ensuring that most of the DNA is entrapped in the bilayers [118]. Liposomes in vaccine formulations can regulate the local tissue distribution, retention, and cell trafficking at the injection site [119]. Thus, some of the advantage of vaccines 430 prepared using liposomes are their induced enhanced antibody production and CTL responses. However, the features of immune responses enhanced by liposomes are different due to the lipid composition, particle size, surface charge, and the location of antigens to be entrapped [110]. Studies have demonstrated that liposomes show promise as a candidate for delivery of DNA vaccine to mucosal tissue.”

In my opinion the title does not represent exactly the provided information in the manuscript. The first part of the manuscript is about different types of vaccines. And the title could be “..vaccines and strategies for pandemic preparedness …”

Section 2. Avian influenza vaccines

Provided information is not equal for all parts.

2.3. Virus-like particle (VLP) vaccines – you could add an example of these type of vaccine.

Section 3. Strategies for DNA vaccine delivery

Well written but no examples of avian influenza vaccines and their effectiveness.

Section 4. DC-targeting enhances vaccine immune efficiency

It is better to write the whole name in the title of the section, not abbreviation, i.e. Dendritic cell.

In conclusion, I will say that the data provided is useful, but it is more suitable for a textbook. Examples and discussion should be added on what exactly the strategies are, when vaccinating poultry against avian flu. What is the effectiveness or benefit of the method used? The authors describe the vaccines themselves and the strategic model, rather than showing what the outcome of their application is.

Author Response

Pls see attachment

Round 2

Reviewer 1 Report

The paper by Hai Xu, Shanyuan Zhu, Roshini Govinden and Hafizah Y. Chenia entitled “Multiple vaccines and strategies for pandemic preparedness of avian influenza virus” reviews the literature on different types of vaccines against avian influenza viruses of pandemic potential. The paper can be interesting for a wide range of experts in virology, molecular biology and biotechnology.

Reviewer 2 Report

The manuscript is well written. I agree with the title and the corrections made.